# Anxious, Depressed, and Suicidal: Crisis Narratives in University Student Mental Health and the Need for a Balanced Approach to Student Wellness

**DOI:** 10.3390/ijerph20064859

**Published:** 2023-03-09

**Authors:** Jason Bantjes, Xanthe Hunt, Dan J. Stein

**Affiliations:** 1Alcohol, Tobacco and Other Drug Research Unit, South African Medical Research Council, Cape Town 7505, South Africa; 2Institute for Life Course Health Research, Department of Global Health, Stellenbosch University, Stellenbosch 7602, South Africa; 3Department of Psychiatry and Mental Health, University of Cape Town, Cape Town 7700, South Africa; 4SAMRC Unit on Risk and Resilience in Mental Disorders, Stellenbosch University, Stellenbosch 7602, South Africa

**Keywords:** student mental health, university, college, crisis narratives, depression, anxiety, medicalization

## Abstract

There is growing global awareness of the poor mental health of university students, as well as the need to improve students’ access to services and expand the range of available evidence-based interventions. However, a crisis narrative is emerging, particularly in the wake of the COVID-19 pandemic, that runs the risk of positioning all students as potential patients in need of formal psychiatric interventions. Our aim in this commentary is to critically present the evidence that supports increased attention to student mental health, while also raising a concern that the crisis narrative may itself have unintended harmful consequences. We highlight some of the potential dangers of overtly medicalizing and thus pathologizing students’ experiences of everyday distress, inadequacies of formal diagnostic categories, limitations of focusing narrowly on psychotherapeutic and psychiatric interventions, and the short-sightedness of downplaying key social determinants of students’ distress. We argue for an integrative and balanced public health approach that draws on the rigor of psychiatric epidemiology and the advances that have been made to identify evidence-based interventions for students, while simultaneously being mindful of the shortcomings and potential dangers of working narrowly within the paradigm of diagnostic labels and psychotherapeutic interventions.

## 1. Introduction

In the past decade, a dominant narrative has emerged across the globe about a university student mental health crisis and the need for urgent action, often assumed to entail formal campus-based psychological or psychiatric interventions [1,2,3,4]. This “crisis narrative” has positioned students as anxious, depressed, and suicidal, potentially creating the impression that addressing the mental health of students may be one of the most important global priorities in higher education [5,6]. Although the crisis narrative pre-dated COVID-19, it has gained prominence in the wake of the pandemic because the global response to the emergence of the SARS-CoV-2 virus disrupted students’ academic and social lives and heightened peoples’ sensitivity to vulnerability. The pandemic has also prompted mental health professionals to vociferously reassert the importance of their discipline in a world where infectious disease specialists are foregrounded [7,8,9], resulting in the mental health consequences of the pandemic being touted as the “second pandemic” [10] and a potential “Tsunami” [11]. It is too early to tell how the history of COVID-19, including the intersections of COVID-19 with mental health, will be written, but nonetheless it seems clear that the pandemic has fueled the student mental health crisis narrative [12,13,14,15,16].

Crisis narratives are, of course, not unique to mental health, and they have been the focus of scholarly attention in a range of public health settings, including infectious disease pandemics [17,18,19,20]. At their worst, crisis narratives sow panic and create confusion through use of hyperbolic rhetoric [18]. However, crisis narratives can also serve a useful social function, acting as a sort of ‘balm’ that creates cohesion within an “in group” while also setting up an ‘us versus them’ dialectic [20]. Seeger and Sellnow [17] have asserted that crisis narratives “frame larger public and societal understanding of risks, warnings, and potential harms” (p. 5), thus potentially spurring helpful health promoting action.

In this commentary, we present the empirical evidence that supports increased attention to university student mental health, but we also raise a concern that the crisis narrative may itself have unintended harmful consequences. We argue that the crisis narrative runs the risk of pathologizing students’ everyday experiences of the typical stressors associated with being a young adult, and that to advance student wellness we need a balanced public health approach. The empirical evidence we present is drawn from systematic reviews (where these are available) or other large and/or multinational studies. We also draw on our own experience as researchers working actively in the field of student mental health for more than 10 years. As is customary in commentaries, we have offered our own observations and opinions, based on a critical stance towards our own work and contemporary research trends in this field. We have included citations to link our arguments to broader critiques of mental health research and practice. Where no citations are offered, the ideas expressed are based on our opinions and critical reflections.

## 2. Evidence of the Need for Attention to Student Mental Health

Ample epidemiological evidence supports the need for attention to student mental health, with data showing high rates of mental disorders and suicidality among college students, although much of this data comes from high income countries [21,22,23]. One multinational study across 19 universities in 8 countries (Australia, Belgium, Germany, Mexico, Northern Ireland, South Africa, Spain, and the United States) reported lifetime and 12-month prevalence estimates for common mental disorders of 35% and 31% [24], and 12-month prevalence estimates for suicidal ideation, plan, and attempt of 17.2%, 8.8%, and 1.0%, respectively [25]. The lifetime- and 12-month prevalence of non-suicidal self-injury among students is reported to be as high as 17.7% and 8.4% [26]. Similarly, our own work in South Africa, that consisted of a national survey of students (n = 28,268) from 17 universities, found 30-day prevalence estimates of 16.3% for mood disorders, 37.1% for anxiety disorders [27], and 24.4% for suicidal ideation. Treatment rates are also typically low, with data (which predominantly come from the USA) showing that most students who have mental disorders do not access mental health services [28], and that students face several barriers to treatment including availability, accessibility, affordability, and a range of institutional (i.e., contextual, and structural) and individual (i.e., perceived need for treatment and attitudinal) factors [29]. One multinational study (with data from Australia, Belgium, Germany, Mexico, Northern Ireland, South Africa, Spain, and the United States) found lifetime- and 12-month treatment rates among students were very low, with estimates of 25.3–36.3% for mental disorders and 29.5–36.1% for suicidal thoughts and behaviors [30]. Left untreated, these mental health problems result in profound role impairment [31] and are associated with academic failure [32,33].

The impact of COVID-19 on students’ mental health has been widely documented, with research showing that COVID-19 and the associated educational and governmental mitigation strategies typically had a deleterious impact on students’ mental health [13,14,15,16]. Similarly, a range of social and cultural factors appear to have adversely affected the mental health of young people, including living in a technology rich environment [34], social networks, and changing ideas about parenting and education [35].

There is also a growing awareness that university faculty are no less distressed than students, at least on university campuses in the United States [36]. Reports suggest that the crisis narrative may be spilling into the discourse of mental health on university campuses more broadly and expanding to include educators in higher education [37].

## 3. An Industry of Student Mental Health Research

The increased global interest in student mental health has created an “industry” for mental health researchers, who are typically located in universities and thus have relatively easy access to survey respondents. Student mental health data can be collected (at relatively low costs) using self-report online surveys with students recruited by emails sent to their university email addresses. Indeed, researching the mental health of university students is often (although not always) considerably cheaper and logistically easier than community-based epidemiological projects (especially community-based studies with hard-to-reach populations, such as migrants or illicit drug users, who may be at high risk for mental health problems). The relative ease with which data can be collected from students may result in a flood of research on students’ mental health, while other hard-to-reach populations remain neglected.

Campus-based surveys may not be as expensive as community-based epidemiological studies, but nonetheless the costs are not negligible; researchers who are dependent on securing funding for such work may thus have a perverse incentive to over-state the magnitude of the problem and frame findings in ways that feed the crisis narrative. Although there is no direct evidence of this kind of distortion of evidence (as far as we know) in the field of student mental health, other scholars have documented the existence of “scientific fraud” [38] and noted the complex relationship between the desire to attract research funding and both setting research agendas [39] and knowledge creation processes [40].

Our point here is not that the existence of an industry for student mental health is necessarily “bad”, but rather that careful consideration needs to be given to the potential advantages of improved knowledge of student mental health versus the potential for iatrogenic harmful consequences of an industry focused on this area.

## 4. Benefits of Focusing Attention on Students’ Mental Health

Focusing attention on students’ mental health and advocating for resources to support students has several distinct advantages.

First, late adolescence and early adulthood are periods of peak onset for many common mental disorders [41], making the transition to adulthood (which typically coincides with entry to university) a critical juncture during psychopathology and mental health [42]. More campus-based interventions and student support services could help with early identification and treatment, preventing more serious problems in the future. Second, left untreated student mental health problems have a deleterious impact on social functioning, cause serious role impairments, and increase the risk of suicide [43,43]. Third, mental health is associated with academic performance [32,33], so promoting the mental health of students is a good educational strategy for improving retention and promoting academic attainment.

For all these reasons, drawing attention to an important issue (the wellbeing of young people) and mobilizing resources to support students during what can be a tricky and important developmental period seem worthwhile strategies. We have seen firsthand how this has occurred in South Africa, where findings from the national student mental health survey have precipitated public conversations and provided data to direct evidence-informed responses, which may improve students’ access to support services [27].

## 5. Could There Be Unintended Harmful Consequences of a Crisis Narrative?

Nevertheless, when attention to student mental health is framed in terms of a crisis narrative, it may also have unintended negative consequences. There are at least four potentially harmful consequences of a crisis narrative that is focused on pathologization, that under-emphasizes resilience, and that downplays the importance of typical stressors during early adulthood, as well as the need for well run and reasonably resourced universities.

First, a crisis narrative may inadvertently discourage some students from seeking treatment. When academics or the media report ubiquitously high rates of mental disorders and very low treatment rates, they could create the impression that having these symptoms is normal for students; paradoxically, this could discourage treatment seeking. For example, reporting that 60% of university students are living with mental health conditions but very few of them receive treatment [44] could easily be read as confirmation that depressive symptoms are a normal part of students’ experience and that most students deal with this problem on their own, thus decreasing the likelihood that students will see these as symptoms that warrant professional attention. It is potentially unhelpful to repeatedly tell young people (who are typically very sensitive to peer group norms) that most other students deal with feelings of depression and suicide on their own.

Second, increasing awareness of students’ mental health problems can paradoxically contribute to higher and higher rates of mental illness being reported; this phenomenon has been called the “prevalence inflation hypothesis” [45]. Increasing students’ awareness of very high rates of mental illness through crisis narrative awareness campaigns could lead some students to interpret and report relatively mild forms of distress as mental health problems, rather than interpreting these as the everyday vicissitudes of life [45]. This, in turn, could also promote the idea that students need formal psychological/psychiatric treatments for any emotional distress. This downplays the value of other helpful kinds of intervention and can lead to professional services being so overwhelmed that they are unable to do the work of caring for those who are more seriously ill.

Third, a crisis narrative that overstates the number of students with diagnosable mental disorders runs the risk of trivializing serious mental illness. In her provocative award-winning article (aptly entitled ‘It’s nothing like a broken leg’: why I’m done with the mental health conversation), journalist Hannah Jane Parkinson explains how mental health awareness programs and public conversations about mental health that frame mental illnesses as easily treatable medical conditions run the risk of trivializing serious mental illness [46]. Our argument here, which links to broader issues about the neglect of the seriously mentally ill [47,48], is that student mental health research that frames relatively mild symptoms of distress and everyday stress as pathological can turn attention and resources away from students with serious mental illnesses. It may not make much sense, for example, to group students with mild (non-clinical) symptoms of social anxiety or students who are negotiating a broken heart together with students impaired by serious mental illnesses such as bipolar mood disorder and schizophrenia. There are some symptoms of mental illness during late adolescence that resolve over time with minimal professional intervention and rather with appropriate support from family and friends. Casting the net too widely when defining students’ mental health problems could (especially in resource constrained environments) draw resources and attention away from students with serious mental illnesses who need specialized (often expensive) campus-based services and ongoing professional support to complete their university education. Our own work in South Africa aptly illustrates this point. In a recent national mental health survey, we assessed 11 common mental disorders and found that 71.3% (S.E. = 0.5) of students screened positive for at least one of the disorders in the preceding 30-days [27]. This statistic could easily create the impression that resources are needed for large scale population-level interventions to reach the 71.3% of students with “mental illness”, even though some of these symptoms were very mild and some could resolve without intervention. However, allocating resources to population-level interventions in South African universities would redirect resources away from much needed campus-based tertiary interventions for students with debilitating severe mental illnesses.

Fourth, a crisis narrative can position all students as patients in need of intervention and thus obscure the ways in which many students demonstrate resilience. The rise of therapeutic culture [49] in broader society and the corresponding trend towards “therapeutic education” in schools and universities [50] has created the impression that it is normal to think of all students as psychologically vulnerable and in need of protection and intervention. However, even if rates of mental illness are marked among students, it would be a mistake to assume that the proportion of students with symptoms of mental illness equates with the proportion who want or perceive a need for treatment. For example, in the South African national student mental health survey, slightly less than two-thirds of students (60.5%, S.E. = 0.5) who screened positive for a mental disorder perceived a need for treatment and thus self-identified as patients in need of intervention. The crisis narrative in student mental health positions all students as “patients in waiting” (i.e., people waiting for treatment or who are at high risk of developing symptoms and thus needing treatment) and contributes to the image of students as vulnerable, while obscuring the reality that many students are resilient, doing well, and even thriving. There is a growing body of research on students’ resilience, with one recent systematic review identifying 72 peer-reviewed articles published between 2007 and 2017 [51]. The resilience literature itself has risks; it may at times underplay important psychopathology (not everyone can be resilient) or oversell the efficacy of resilience interventions. A simplistic focus on the value of “grit”, for example, can be a variation of the crisis narrative using the discourse of resilience.

Fifth, the student mental health crisis narrative typically relies on the methods of psychiatric epidemiology and the language of “disorders” and “symptoms”. Such an approach frames students’ difficulties in terms of symptoms and illness; this may run the risk of pathologizing students’ everyday struggles. Viewing ordinary stressors through the lens of trauma and PTSD, for instance, could easily spill over to conflating emotional distress with mental disorders that need specific psychiatric interventions [52], and may contribute to what Patel has called the “credibility gap” in the field of mental health [53]. It is one thing to claim that students experience distress as they negotiate the developmental tasks of young adulthood and balance the academic, social, and financial challenges of being a student. It is something altogether different to assert that students who struggle have mental illnesses that requires specialist interventions from a mental health professional. The medicalization of social injustice is harmful and detracts from the structural causes of distress, drawing attention away from the types of interventions that are needed to achieve equality and social, economic, and political reforms.

Acknowledging that students sometime struggle and may experience distress raises awareness of the need for a broad range of interventions, including mobilizing personal resources and social support. By comparison, asserting that many (if not most) students are suffering from mental illnesses suggests that drug and psychological interventions are widely indicated. Furthermore, pathologizing the struggles that students face can easily minimize the key social determinants of students’ distress, such as discrimination and marginalization, food and housing insecurity, poor secondary schooling, and inadequate preparation for university. There are clear links here to “psychiatrization” (i.e., the expanding reach of psychiatry and expansions in the number and inclusiveness of psychiatric diagnoses that can lead to over-diagnosis, over-treatment, and over-prescription) [54,55], and “concept-creep” (i.e., the gradual expansion of the meaning of harm-related concepts) [56], which have implications for student wellness research and practice.

## 6. The Need for a Balanced Public Mental Health Approach

Accounts of psychiatry (as with accounts of medicine more broadly) invariably run the risk of being either hagiographic or overly critical. A hagiographic account of the focus on student mental health might emphasize the considerable advances that have been made to scrupulously document the epidemiology of mental illness among students; quantify the need, scale and scope of interventions; and identify high risk groups for targeted interventions and develop and rigorously evaluate a range of psychotherapeutic individual, group, and digital interventions. On the other hand, a more critical account might point to the dangers of overtly medicalizing and thus pathologizing students’ experiences of everyday distress; the inadequacies of formal diagnostic categories; the limitations of focusing narrowly on psychotherapeutic and psychiatric interventions; and the shortsightedness of downplaying key social determinants of students’ distress [57]. We argue that both these extreme positions are over-simplistic and unhelpful. Instead, what is needed is a more nuanced integrative public health approach that draws on the rigor of psychiatric epidemiology and the advances that have been made to identify evidence-based interventions for students, while simultaneously being mindful of the shortcomings and potential dangers of working narrowly within the paradigm of diagnostic labels and psychotherapeutic interventions. There are several distinct areas in which a public mental health approach to student wellness could help to restore balance to the crisis narrative.

### 6.1. Move beyond Diagnostic Labels in Student Mental Health Research

As noted earlier, the measures used to assess mental health in much of the student mental health research are typically based on contemporary western psychiatric diagnostic classifications, including the American Psychiatric Associations Diagnostic and Statistical Manual of Mental Disorders (DSM). The DSM has, for a long time, faced criticisms [58] that may be relevant in the field of student mental health. For one thing, concern has been expressed about the indiscriminate applicability of the DSM’s nosological system in different cultures [59]. Furthermore, mental disorders fall on a continuum, ranging from the experience of one or more symptoms of a mental disorder to a cluster of symptoms that may or may not meet full diagnostic criteria [60], highlighting the meaninglessness of the idea of a clinical cut-off point to clearly differentiate “illness” from “health”. Crucially, classification systems such as the DSM are used in clinical settings to diagnose mental disorders and to inform treatment. However, when applied at a population level to characterize student wellness, the use of clinical diagnostic categories may not facilitate appreciation of the key points that, during adolescence, there are ordinarily high rates of distress associated with this life-stage. Thus, there is a high risk of over-diagnosis alongside an assumption that the large numbers of students who screen positive need formal psychiatric or psychotherapeutic treatment. One possible alternative to using psychiatric screening tools would be to conduct research that explicitly asks students what they are struggling with, what help they need, and how they would like this help provided.

### 6.2. Consider Ways in Which Students Demonstrate Resilience

One alternative to endless research that positions all students as vulnerable and in need of psychiatric treatment is research documenting the ways in which students demonstrate resilience, with the starting assumption being that they are already resilient. We acknowledge that there may be methodological and ideological problems with the concept of resilience and that work may still be required to define and operationalize what is meant by resilience in the context of higher education, as already noted by other scholars [53]. Nonetheless, the point here is that while it is useful to have research within a psychiatric paradigm, there is also potentially merit in research within the paradigm of positive psychology and other strength-based approaches to student wellness. One clear example of this is the ongoing multinational study into character strengths and flourishing that attempts to move beyond the concept of resilience [61]. More recent iterations of resilience and strength-based approaches avoid some of the pitfalls of earlier work and could potentially be helpful for a balanced approach to student wellness [62].

### 6.3. Document Level of Impairment (Not Just Symptoms) and Serious Mental Illnesses

Student mental health research typically focuses narrowly on symptoms of depression, anxiety, and suicidal ideation, and determines prevalence estimates using a cut-point on a screening instrument, such as the Patient Health Questionnaire-9 (PHQ-9) and the Generalized Anxiety Disorder 7-item questionnaire (GAD-7). There is growing appreciation for the idea of a mental health continuum with no clear boundaries and with considerable variations in symptom intensity, frequency, and associated level of impairment [63]. For example, even if 30% of students screen positive for depression, there will be considerable variations in the severity of symptoms and the level of impairment among those who screen positive. It would thus be helpful if student mental health research documented the level of impairment associated with symptoms and focused on assessing severe mental illnesses.

### 6.4. Embrace Etiological Complexity and Pluralism

Identifying risk factors for student mental health problems is important for planning targeted interventions. Similarly, understanding causal mechanisms will help inform primary prevention measures. However, there is a need to guard against overly simplistic linear explanations for the etiology of students’ psychological distress, and to hold the tension between neuroreductionism and culturalism. Neuroreductionism will lead to over simplistic scientistic formulations for students’ mental distress that are devoid of both context and situated meaning. However, the opposite, culturalism, suggests that mental illness is entirely due to social determinants and that if only we address factors such as social inequity, all psychological distress will disappear. Instead, a balanced public health approach to student wellness explicitly acknowledges that mental health is shaped by a range of risk and resilience factors spanning the biopsychosocial spectrum. By embracing etiological complexity, a public mental health approach to student wellness would advocate for a pluralistic approach to intervention with a wide variety of different solutions and strategies.

### 6.5. Adopt a Life-Course Approach to Promoting Student Mental Health

It is important to understand how developmental processes interact with social determinants, contextual factors, and biological processes to precipitate and maintain students’ psychological distress. A life-course approach to student mental health acknowledges that some of the suffering students experience is due to negotiation of the typical developmental tasks of young adulthood (e.g., formation of self-identity, development of romantic relationships, learning to tolerate frustration and regulate emotions) [64].

### 6.6. A Wide Array of Interventions

One approach would be to expand professional counselling services and increase students’ access to evidence-based individual and group interventions. This seems appropriate, given the prevalence of mental disorders reported among students and the evidence supporting the effectiveness of such interventions [65]. However, a public mental health approach would go beyond this to emphasize the need for campus-wide (i.e., eco-systemic) interventions to address social and contextual determinants of well-being. A public health approach would also explore how task sharing models (including non-professional and peer-to-peer interventions) may help to scale up access to psycho-social support, although such approaches would need to be implemented with good supervision, adequate training, strategies to ensure competencies of those offering support, and appropriate ongoing monitoring and evaluation. Beyond this, there is a need for universities to have sound policies and practices that support student wellbeing, including, for example, a mental health policy and policies to address harassment and bullying. Perhaps more important than any specific intervention is being willing to commit to a process of measurement and learning and of building community. This process of measurement and learning may be based in a public health model; however, it requires not only a consideration of the principles of this field, but also of the particulars of any specific institution. A broad array of targets may be relevant at any one university, ranging from improvement of wellness services, through a revision of pedagogical practices to minimize bullying and to a range of ways of building community. Finally, universities should develop interventions that focus on problems that are important to students (i.e., self-identified causes of distress) rather than just interventions that seek to reduce symptoms of common mental disorders.

### 6.7. Responsible Messaging

Researchers and the media need to be thoughtful about the implications of framing student mental health using a crisis narrative and inadvertently trivializing serious mental illnesses and/or creating the impression that mental illness is normal for students. For example, when reporting prevalence estimates for subclinical symptoms, it should be very clear to lay audiences that these are not necessarily the same as the proportion of students with clinically significant symptoms requiring professional intervention. Likewise, acknowledging confidence intervals when reporting prevalence estimates is important; stressing the lower bound of a confidence interval may help lay audiences appreciate that the prevalence may be lower than it appears. Take for example, a recently published meta-analysis reporting a pooled prevalence of suicide-related outcomes among undergraduate students of 14% (95% CI 0%, 44%) [66]. Presenting these findings to the public as “14% of students are suicidal” is different from the (perhaps less sensational) reporting that the number of students who are suicidal may be as low as 0% but may also be higher.

Rather than simply reporting sensationally high prevalence estimates of psychological distress and encouraging all students who experience such distress to seek professional help, we need to emphasize that a range of interventions are useful for addressing the normal stressors of university life (including developing healthy relationships with peers and mentors), and that more serious mental disorders are responsive to professional intervention.

While many of the arguments we make here focus exclusively on the mental health of university students, many of the points we have highlighted and the suggestions we have made could apply to discourses about the mental health of adolescents and young people more broadly.

## 7. Conclusions

There can be little doubt from the available data that students experience mental health problems that are sometimes debilitating and that interventions are needed to help support the wellbeing of students. We need reliable and meaningful epidemiological data alongside advocating for resources to expand students access to evidence based interventions globally. However, we also need to be aware of the consequences of creating or perpetuating a crisis narrative that frames all students as patients in waiting. Clearly there is a need to advocate for increased resources for students with mental health problems and this may require the attention of university administrators, funders, and the media. However, framing student mental health as a crisis may have several unintended and potentially harmful consequences, including medicalizing students’ everyday experiences, conflating distress and disorder, and inadvertently implying that students are all potential patients in need of psychiatric interventions. A balanced public health approach to student mental health may help draw attention to the issue of students’ need for support and guidance, while avoiding inappropriate pathologization. Such an approach emphasizes the need to embrace complexity and to advocate for pluralism in student mental health research and interventions.

## Data Availability

Not applicable.

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
