# Peer review of "Anxious, Depressed, and Suicidal: Crisis Narratives in University Student Mental Health and the Need for a Balanced Approach to Student Wellness"

_ijerph, 2023, doi:10.3390/ijerph20064859_

Round 1
Reviewer 1 Report
Peer-review of ' Anxious, depressed, and suicidal: Crisis narratives in student mental health and the need for a balanced approach to student 3 wellness’ (in IJERPH, manuscript # 2269387)
General Comments
Commentary submissions for IJERPH should be thought-provoking viewpoints with a peppering of opinion blended into strongly supportive referencing, all pointing toward potential research avenues, policy implications, and/or further discourse—the authors have exceled in the task. The Commentary is an elegantly written, well-balanced manuscript that is worthy of publication and widespread dissemination.
Even though the manuscript qualifies for publication as is, it is the opinion of this reviewer that with some minor revisions and tightening, this could be a stand-out article for IJERPH, and worthy of media attention.
Suggested Revisions
1. Please add at least a few lines early on in the manuscript with a reference or two on the term crisis narrative(s). The term is approached tangentially on line 36 with quotation marks over the term. Readers might appreciate a few sentences related to scholarly discourse on crisis narratives. Some scholars suggest that crisis narratives serve a social function, a sort of ‘balm’ that brings together an ‘in-group’ in an ‘us vs. them’ way; crisis narratives are not unique to mental health and there is quite a bit of scholarly discussion on how such narratives, as Matthew Seeger and Timothy Sellnow state in “Narratives of Crisis: Telling Stories of Ruin and Renewal” (Stanford University Press, 2016. pg. 5) “frame larger public and societal understanding of risks, warnings, and potential harms.” Since the term crisis narratives is in the title of the manuscript, and mentioned close to 2 dozen times throughout, it should be expanded upon early on in the manuscript. See also
Glowacki EM, Taylor MA. Health hyperbolism: A study in health crisis rhetoric. Qualitative Health Research. 2020 Oct;30(12):1953-64.
2. I would recommend removing “Fortune” magazine as a source and replace it with scientific journal articles. Here is a reference that should be added.
Lipson, S.K., Zhou, S., Abelson, S., Heinze, J., Jirsa M., Morigney, J., et al. (2022). Trends in college student mental health and help-seeking by race/ethnicity: Findings from the national healthy minds study, 2013-2021. Journal of Affective Disorders, 306, 138-147.
3. The authors underscore that overly simplistic linear explanations for students psychological distress should be avoided. This is an important point. However, it might be worth pointing out that reported increases in mental disorders/distress appear to be culturally driven. Doing so will add to the legitimacy of the primary argument made by the authors. See
Twenge, J.M. (2020). Why increases in adolescent depression may be linked to the technological environment. Current Opinion in Psychology, 32, 89-94.
Dupont S, Mikolajczak M, Roskam I. The cult of the child: A critical examination of its consequences on parents, teachers and children. Social Sciences. 2022 Mar 21;11(3):141.
4. The title of the article uses the word students which could be interpreted as all youth and post-secondary school attendees of all ages. The article focuses almost exclusively on university students. Either the title should be changed to specify ‘university students’ or some level of clarification should be provided in the body of the manuscript. Much of the crisis narratives on youth mental health is centered around adolescents. On this topic, what about the less-youthful faculty and educators in these environments? Research shows that faculty are no less distressed on American university campuses, so to what extent is the crisis narrative spilling into educator mental health? See
Meeks, K., Sutton Peak, A., Dreihaus, A. (2021) Depression, anxiety, and stress among students, faculty, and staff. Journal of American College Health, DOI: 10.1080/07448481.2021.1891913.
5. In Section 6.1 the authors describe the problematic nature of the DSM-5 (etc.). Please add some references here. There are many critiques of the DSM and its problems. This might be a good point to add a line or two about westernization and medicalization. It is also an opportunity to add a line about the ongoing multinational study into character strengths and flourishing (beyond resilience per se). See
Abramov DM, Peixoto PD. Does contemporary Western culture play a role in mental disorders? Frontiers in Psychiatry. 2022 Sep 7:2055.
Merlo G, Vela A. Mental health in lifestyle medicine: a call to action. American Journal of Lifestyle Medicine. 2022 Jan;16(1):7-20.
VanderWeele TJ. The importance, opportunities, and challenges of empirically assessing character for the promotion of flourishing. Journal of Education. 2022 Apr;202(2):170-80.
Author Response
Thank you for the positive and affirming feedback on our manuscript. We are also very grateful for your constructive comments and suggestions – we found these very helpful in refining our submission.
We have responded to your feedback by making the following changes:
- We have added a short paragraph within the introduction to say a bit more about “crisis narratives” and the use of this construct more broadly. Thank you very much for the references (we had not seen these) but they have led us down a path of thinking more carefully about the use of rhetoric in framing student mental health. We have added citations to make these links clear to our readers.
- We have removed the reference to Fortune magazine and added the citation you referred us to. Thank you for pointing out this oversight.
- We have included the point you make about socio-cultural drivers of rising mental health problems in young people, and referenced the work you pointed us towards. Again, thank you for pointing out our blind-spot here.
- Your point about the social and cultural context, and its effects on youth mental health is well made. We have amended the text to reflect this and added the citations you suggested.
- We have amended the title to reflect your comments about the manuscript’s focus on the mental health of university students. We have also briefly acknowledged the literature on the mental health of faculty in higher education settings, and cited the paper you suggested. We have also included your point (and suggested reference) about the mental health of faculty. In the conclusion we have also acknowledged that many of the arguments we make are not unique to university students and may apply to discourses about the mental health of adolescents.
- Thank you for the references about critiques of the DSM and the literature on flourishing. We have added these to section 6.1 and section 6.2. we have also fleshed out some of the concerns about using DSM type western psychiatric systems, and added citations for these ideas.
Thank you again for the very helpful and constructive comments. They have been invaluable.
Reviewer 2 Report
This well-written and well-balanced paper comes at an opportune time though it tackles long-standing issues. I hope that this paper catalyzes much needed discussions about how we address mental health topics, in higher education and more generally.
A quick note, there might be some text errors that should be double-checked. For example, line 93 has "versus intended harmful consequences" where I assume 'unintended' or iatrogenic was what should have been there.
Author Response
Thank you for the very affirming and positive feedback on our manuscript.
We have corrected the error in our use of the word “intended” (which should have been iatrogenic), and carefully read the manuscript for other typos.
Reviewer 3 Report
1. Since the crisis narratives related to mental health in this study are defined in the context of Covid-19, I suggest you incorporate covid-19 in the topic.
2. It is important to introduce a methods section that details how the commentary was written and/or scholarly conducted. One of the key issues that should be addressed in the section is scholarly processes a footed by the authors for the evidence gathering strategies employed. Also, what kinds of evidence were gathered? What makes them valid and legitimate documents (epidemiological evidence, etc.) ? What criteria governed their selection? How were they analyse in a rigorous and scholarly manner? What time-tested analytical procedure(s) were adopted and why? The Methods section must discuss these matters.
3. It will be relevant to mention the countries involved in the multinational studies discussed.
4. While the arguments presented in the section, 'An industry of student mental health research' are interesting, they are highly opinionated (though the paper is a commentary), many of the wild claims need to be substantiated with scholarly evidence. For instance, what studies have confirmed the deliberate exaggeration of findings by researchers in lieu of gaining the favor of their funders? I also doubt and contest the generic expression that campus-based surveys are relatively cheaper making it a purposive 'industry' for mental health researchers. The tone needs to be circumspect as this is a scholarly writing with evidence buttrressing the assertions made.
5. In the section, 'the benefits of focusing attention on students'mental health', kindly reference the South African Experience presented and highlight the key findings from that study upon which the conclusion presented in the section has been made.
6. The arguments in the section 'Could there be unintended harmful consequences of a crisis narrative?' are interesting. Yet, in the first position, don't you think failure to objectively report the results of a scientific study, with the projected implications on students attitudes toward mental health alluded by the authors in mind, could also jeopardize scientific production and draw up way from scientific truth? The first point needs to be re-written with this in mind.
Secondly, the increasing awareness on the need to report mental health among the student population should be seen as a good omen to arrest all forms of mental health akin to the age categories as affirmed by the authors. The other side of the coin or the opposite suggested as a strength by the authors could also have tremendous consequences for future mental health crisis among university student population?
The third, fourth and fifth arguments are scholarly and academically stimulating, offering enough scholarly references and arguments that are well substantiated.
7. The greatest strength and novelty of this commentary has to do with the seven suggested ways of advocating a balanced public mental health discourse rather than following the usual narrative and approaches used in arresting mental health cases and studies.
Author Response
Thank you very much for your very useful comments and the careful and critical reading of our work. We appreciate your guidance in helping to make this commentary more scientific and our argument more rigorous. Please see below a point-by-point response to your feedback:
- Thank you for pointing out the problem with our highlighting of COVID-19. We can see how this created the impression that this commentary was directly related to the pandemic. What we had intended instead was that COVID would serve as an example of a recent cultural event that has exacerbated problems with student mental health. We have corrected this now by also adding examples of other contemporary socio-cultural events that may have contributed to the crisis narrative in student mental health.
- We understand your concern about the need to be explicit about the approach we used (i.e. method) and can see value with this. we have however consulted with the editorial staff of the journal who have confirmed that a specific subsection for “Methods” is not appropriate for a commentary. We have thus retained the convention of not having a methods section. We have however added some more detail about the approach we used and our subject position. Please see the sentences added at the end of the introduction explaining our approach and subject position.
- We have added relevant detail to clarify where data were collected in multinational studies.
- We take the reviewer’s point about the highly opinionated comments in section 3 (“An industry of student mental health research”). We have re-worked this section and added more citations to support the argument we are making. As is customary in commentaries, we have expressed our own opinions and observations, but we have amended the manuscript to make it explicit that were no citations are given the views expressed are solely the opinions of the authors. We also understand that not all student mental health research is inexpensive – but we believe that this point is clear now in the amended text. Thank you for helping us to make the manuscript more rigorous and “scientific”.
- We have added a reference for the South African experience included in section 4 of the manuscript.
- We understand the reviewers concern about the speculative nature of our comments in section 5 (“Could there be unintended harmful consequences of a crisis narrative?”), particularly in relation to our first and second points. We reiterate that where no citations are included the opinions expressed are those of the authors based on our observations. We have made this clear in the introduction. Also, our use of words like “can” and “could” in this section explicitly signal to readers that we are not making truth claims here but rather speculating.
Thank you for helping us to refine this work and to ensure that we have been responsible in the way we speculate about the possible iatrogenic effects of a crisis narrative in student mental health.
Round 2
Reviewer 3 Report
Thank you for addressing all the issues I raised and I support your position detailed in the revised manuscript and author response letter. Regards